# Comparison of Named Data Networking Mobility Methodology in a Merged Cloud Internet of Things and Artificial Intelligence Environment

**DOI:** 10.3390/s22176668

**Published:** 2022-09-03

**Authors:** Wan Muhd Hazwan Azamuddin, Azana Hafizah Mohd Aman, Rosilah Hassan, Norhisham Mansor

**Affiliations:** 1Center for Cyber Security, Faculty of Information Science & Technology, Universiti Kebangsaan Malaysia, Bangi 43600, Malaysia; 2Department of Electrical Technology, Advance Technology Training Center (ADTEC) Batu Pahat, Batu Pahat 83000, Malaysia

**Keywords:** artificial intelligence, cloud internet of things, named data networking, network analysis model

## Abstract

In-network caching has evolved into a new paradigm, paving the way for the creation of Named Data Networking (NDN). Rather than simply being typical Internet technology, NDN serves a range of functions, with a focus on consumer-driven network architecture. The NDN design has been proposed as a method for replacing Internet Protocol (IP) addresses with identified content. This study adds to current research on NDN, artificial intelligence (AI), cloud computing, and the Internet of Things (IoT). The core contribution of this paper is the merging of cloud IoT (C-IoT) and NDN-AI-IoT. To be precise, this study provides possible methodological and parameter explanations of the technologies via three methods: KITE, a producer mobility support scheme (PMSS), and hybrid network mobility (hybrid NeMO). KITE uses the indirection method to transmit content using simple NDN communication; the PMSS improves producer operation by reducing handover latency; and hybrid NeMO provides a binding information table to replace the base function of forwarding information. This study also describes mathematical equations for signaling cost and handover latency. Using the network simulator ndnSIM NS-3, this study highlights producer mobility operation. Mathematical equations for each methodology are developed based on the mobility scenario to measure handover latency and signaling cost. The results show that the efficiency of signaling cost for hybrid NeMO is approximately 4% better than that of KITE and the PMSS, while the handover latency for hybrid NeMO is 46% lower than that of KITE and approximately 60% lower than that of the PMSS.

## 1. Introduction

Named data networking (NDN) is a new type of architecture based on the Internet Protocol (IP) architecture that makes the Internet work [1]. To connect the various elements of the network, the IP is used as the network layer, incorporating security and device-to-device connectivity. In Internet hourglass architecture, the thin part is at the center of the universal IP network layer. Recent NDN research adopted an adaptive forwarding strategy based on deep reinforcement learning, maintaining the right data delivery equilibrium and allowing users to pick and compete [2]. With its advantages of speed and reliability, NDN is a highly appealing networking solution for Internet of Things (IoT) applications. The goal of the IoT is to connect everyone and everything at any time and from any location. The IoT is being adopted by an increasing number of users in enterprises and academic sectors [3]. Many issues arise as a result of the rapid expansion of IoT devices on the Internet because the current Internet was not designed to handle resource-constrained devices [4]. However, due to the peculiarity of this new method, a new paradigm for IoT must go a step further [5]. This situation means that an alternative Internet design must be investigated—one that takes the restrictions and quantity of devices into account. Other research indicated that NDN is a crucial enabler of network and compute convergence, upon the edge of which artificial intelligence (AI) should be built [6].

In NDN, a consumer sends an interest packet (I_packet) with a content prefix (prefix) [7]. The router records the interest packet’s interface in the pending interest table (PIT) and looks up the forwarding information base (FIB) for information about the content filled by the name-based routing (NBR) protocol. Some routers or content providers send back information to a consumer and store the content in the content store (CS).

Another aspect of NDN that still has no updated solution is mobility. A survey paper [8] derives aspects of mobility that currently remain unresolved. This study discusses NDN mobility methodologies and measures the performance evaluation for each producer mobility operation: KITE [9], a producer mobility support scheme (PMSS) [10], and hybrid network mobility (hybrid NeMO) [11].

**Motivation:** NDN technology comes with a new perspective as a blank-slate Internet architecture under Information-Centric Networking (ICN). The major goal of NDN technology is to address the existing Internet’s flaws by shifting the IP-based nature of communication to content-transmission communication. Furthermore, NDN was created to facilitate mobile content transmission. Nevertheless, with the rapid development of networking and the proliferation of mobile devices, numerous academics, academic institutions, and organizations have explored ways to manage the mobility of network-connected mobile devices. Moreover, NDN has emerged as the future Internet method of addressing mobility issues. Mobility support enables mobile devices to move between various points of attachment (PoAs) without interfering with content delivery and with minimal handover delay. A PoA makes it possible for mobile nodes to connect to a network. Therefore, mobility was separated into mobility for consumers and producers. According to Saxena et al. [12], the consumer-driven nature of NDN is naturally supported by consumer mobility. When a mobile user moves to a new PoA, the I_packets must be present for transmission to continue normally or to resume after handoff. The mobility of content producers has minimal issues when compared with those of IP architecture [13].

Producer mobility means that a content provider can move without causing problems for consumers and intermediate routers, in terms of content availability and location, with as little time as possible between moves. Consequently, on the basis of a new perspective on mobility support for NDN, Zhu et al. determined that NDN does not support producer mobility [14]. NDN faces a routing table size scaling issue. In addition, when providers relocate to a different location, the naming structure introduces significant scalability challenges [15]. In the same way, when an object moves, a new route must be announced and spread to replace the old routing information.

Another problem that needs to be solved is the long handoff latency and unnecessary I_packet losses during transmission to a producer’s old location [16,17]. When a producer moves, the I_packets keep following the prefix trace in the FIB, which means the I_packets do not reach the producer and are dropped. In addition, indirect points to support producer mobility stretch the data path and cause a problem. Therefore, the circumstances cause a long handoff time. Furthermore, producer mobility requires additional research to find a concrete solution so that the NDN architecture can confidently be used to replace the current Internet architecture, with a lack of problems.

In conclusion, a great deal of optimism has been expressed about NDN becoming a future Internet architecture that is capable of incorporating other networks, without the need for additional mechanisms. Cisco proposed incorporating hybrid ICN into the 5G network [18], which prompted many researchers to improve the transmission of ICN into the 5G network, NDN-5G-SDN support [19], and IoT networking [20,21]. Therefore, more research is needed, especially when it comes to how NDN helps producers move around.

**Article Contribution:** No recent review papers address the latest NDN mobility issues. Previously, most NDN mobility assessments concentrated on NDN surface design and quality, while ignoring NDN mobility methods and unresolved difficulties. This study provides solutions to some of the questions and problems that remain about NDN mobility. This study’s contributions are as follows: (1) It examines NDN technology, AI, cloud IoT (C-IoT), and NDN-AI-IoT; (2) It provides an explanatory methodological and parameter setup for KITE, the PMSS, and hybrid NeMO; (3) It constructs mathematical equations for signaling cost and handover latency; and (4) It conducts a simulation experiment by using NS3 for network performance analysis.

**Article Organization:** The study consists of four sections. Section 2 describes NDN, NDN with AI, and NDN with C-IoT, in terms of features and technology applications. Section 3 describes and compares the mobile operation methodology on mobile producer operations that are integrated with AI and cloud services, i.e., KITE, the PMSS, and hybrid operations. Section 4 provides a performance comparison, including the mathematical equations for the NDN-AI-C-IoT mobility method based on signaling cost and handover latency. Section 5 discusses the network performance for signaling cost and handover latency. Section 6 provides the conclusions and recommendations for future research.

## 2. Background

This section briefly explains NDN technology, NDN with AI implementation technology, NDN with embedded cloud technology, and NDN with IoT application. The current Internet architecture follows the process of data transmission by referring to the open systems communications (OSI) model, but new evolving data-centric technology uses the data-centric NDN model to transmit content, as depicted in Figure 1. NDN transmission focuses on layer 3 for content transmission, while for the OSI layer, the IP packet is used in layer 3. A new NDN paradigm uses a centralized cloud-based strategy for deploying AI techniques. Recently, NDN-AI has been claimed to be one of the primary enablers of network and computer convergence. Other technologies include NDN embedded with IoT technology, such as NDN C-IoT, as a new IoT technology that is aimed at fast data retrieval. To obtain data, the IoT normally uses IP-based data delivery techniques. However, using IP-based ways to achieve NDN C-IoT is difficult, because it introduces a new data-centric paradigm with a distinct design and operation. Current IoT technology suffers from a severe cascading bottleneck that affects network performance; the solution to this issue was provided by Fu and Yang [22]. This study focuses on NDN-IoT to tackle issues that have not been solved by current IoT applications.

### 2.1. Named Data Networking

NDN presents a high-level overview of the concept and associated approaches, in which data are identified based on substance or content, rather than on geographical location. Distributed computing is in high demand, as new technologies, such as big data, data mining, and AI, emerge. Increasing numbers of articles are devoted to NDN. The next-generation Internet is built on the foundation of NDN, which is a typical representation and implementation of content-centric networking (CCN) [12]. A new architecture that can allow large-scale deployment of content-based distribution is needed with the increasing use of the Internet, which has shown the limitations of the TCP/IP protocol stack. NDN has emerged as a game-changing alternative approach and a possible option for next-generation Internet architecture. NDN combines several aspects for routing, security, and mobility.

As far as we know, very little research has considered multi-metric limitations on routing techniques. Several methods have been proposed for routing NDN, one of which is the ant colony optimization algorithm (ACO). The ACO has a better function than other traditional NDN routing techniques, in terms of bandwidth, cost, and delay [23]. As shown in Figure 2 the forwarding strategy (FS) is part of routing in NDN, to process content transmission in NDN communications. Another method, known as ant-colony-based extensible forwarding, was upgraded from the ACO strategy; it enhances sending I_packets by automatically assigning each attribute’s weight [24].

FS has been enhanced to improve quality of service (QoS) and to support data-centric networking in NDN [25]. This approach improved performance significantly. In a dynamically evolving network, current fixed rules have been set up inaccurately. FS, based on Q-learning, has implemented a continuous and online learning method that ensures a rapid response to network outages [26]. Caching strategies also provide several improvements from the previous strategy, using the leave copy everywhere (LCE) strategy. The NDN FS strategy has a function in which content is stored in the cache. However, the LCE strategy uses an algorithm that compares two levels of cache nodes and then takes the node degree into account, decreasing the average number of hit hops, the frequency of cache replenishment, and the throughput of content hits [27].

Another aspect of NDN that is being discussed in current research is handover performance. Network mobility (NeMO) [28] has solved the mobility issues on NDN, but no detailed discussion is available on evaluation performance, i.e., signaling cost and handover latency. Other methods have been proposed, e.g., using IPv6 on mobility and applying SINEMO architecture [29]. Although some problems with the NeMO method have been solved, problems such as low efficiency and delay caused by triangular routing, high handoff cost and latency, and a high level of packet droplet and signaling overhead have not been addressed.

### 2.2. Named Data Networking and Artificial Intelligence

The current TCP/IP network’s security, mobility, and other issues are addressed by NDN, which serves as a prototype for the future network by replacing the IP address with the name of the content. NDN is considered a promising new Internet paradigm to replace the TCP/IP hourglass model’s “thin waist.” A new solution has been obtained by merging NDN technology with AI. With its low power consumption and rapid speed, AI is ideal for IoT systems that require local real-time data analysis. Software implementation of noise distribution normalization [5] adds Gaussian distributed noise, resulting in differential privacy for edge AI. Traffic across the core network and inference latency can be reduced by deploying AI at the edge of networks that are in close enough proximity to the extraction of a large amount of information [30]. Several articles have embedded medical application systems on the IoT, including smartwatches, and smart HVAC systems, with the aim of reducing human effort [31]. A brief discussion was presented by Aman et al. [32], elaborating on the use of the Internet of Medical Things using AI technology. Analyzing local data in real time for the IoT systems is made possible by edge AI, which allows for a low-level operation while protecting user privacy. AI technology has improved data privacy access, based on a large number of data and mobility systems on NDN [33].

In the context of continuing network location changes, mobility refers to the capacity to maintain a unified connection with network entities in AI. Existing host-dimension solutions have exhausted all of the design options and are at a standstill when it comes to new performance advances. Trajectory-driven reachability updating and topology-driven intermediate placement have been proposed to improve overall mobility assistance performance [34]. Figure 3 shows an example of an NDN application that is embedded with AI mobility in e-health technology.

In terms of security, NDN data caching, which is a significant aspect of the technology, can be greatly reduced. Identifying a viable mitigation strategy when NDN routers are exploited by attackers is difficult. ACO [35] has been proposed for probing a safety transmission path and adjusting content retrieval, gathering information about all routers on a network, and bypassing bad routers, thereby preventing the bad routers from continuing to disseminate illegal content and minimizing substance contamination by removing bogus data packets from a cache store during the path traversal. A reactive attack uses content fetch time to track down the items that people are looking for. A proactive attack [36] caches an item before checking whether the victim requests that item.

### 2.3. Named Data Networking and Cloud Internet of Things

Cloud services are increasingly used in today’s networked and distributed applications. However, relying only on cloud services is not always the best option. This section focuses on cloud IoT applications that collaborate with NDN. As a new paradigm technology, NDN has evolved as an element of the future Internet, enabling content delivery, mobile, privacy, reliability, and access to data, regardless of geographic space. NDN has various functionalities, i.e., cache management, device-naming schemes, regulations for password protection, forwarding strategies, configuration management, and awareness of NDN/CCN in the IoT [37]. NDN has also evolved in various IoT applications. The IoT is a well-known idea that envisions the interconnection of several physical items to facilitate the transmission of data between those items. IoT networks need to be designed for performance in a heterogeneous environment with a high volume of data, for devices with limited resources, and with a high degree of mobility. NDN-based content-forwarding solutions have been proposed for usage in IoT applications in NDN-IoT scenarios, given proper identification [38].

NDN IoT applications have also been used to ensure that rapid disease spread can be handled more efficiently, using IoT applications for e-health. To address the limited resources of IoT devices and the time-sensitive nature of the data transmitted, the smart COVID-19 pandemic controlled eradication over NDN-IoT (SPICE-IT) method has been used to reduce the problem of network congestion and cache overflow [39]. The NDN secure remote health-monitoring system [40] is one of the e-health IoT applications that have been developed to increase online communication among doctors and patients. An e-health security application has been deployed to make sure NDN communication is secure [41]; this technology must be enhanced via NDN mobility technology. NDN IoT applications have also been applied in emergency response cases. Currently, all disaster scenarios apply in simulation-based situations only, but an NDN disaster-response system [42] has been developed to monitor the probability of a disaster. That work [42] came up with a new solution by applying edge infrastructure with cloud technology to achieve a fast response for the handover process, ensure a quick exchange of information, and reduce delays in transmission. Another work in disaster scenarios devised a new solution by implementing a connection-oriented surveillance system [43]. That work [43] improved the communication between the sender and the receiver by enhancing the transmission rate of interest packets requested by the consumer.

New NDN ideas, such as those pertaining to security, in-network caching, hierarchical naming or namespace, named content, and NBR, are used to meet IoT needs [38]. Some of the ideas being proposed in IoT technology [44] by using clustering-based intrusion detection for 5G have improved performance on IoT networks. Other aspects of energy-saving on 5G, especially the computing paradigm on green IoT, have been proposed by Hu et al. to improve communication overhead on the server cloud side [45]. This technique can be used for applications on the NDN C-IoT service. NDN is also compatible with 6G networks, because the sender and receiver are connected to each other. NDN has additional features, such as in-network caching and hop-by-hop transmission. Hu et al. proposed an efficient method that is suitable for NDN C-IoT implementation to improve energy in network computing, to overcome higher latency and signaling costs [45].

Over NDN, hydra [46] automatically provides scalable and efficient data distribution at large volumes and enables distributed control with great robustness. The traditional approach is commonly used in the majority of NDN clouds, in which developers need to communicate with a central server. GitSync [47] can function even in the event of a network partition or inconsistent connectivity, because it has no single point of failure. On the other side of the vehicular perspective, vehicle-to-everything (V2X) networking’s full potential can be realized, with several significant issues that must be resolved. The dual technique of a combined blockchain and NDN framework [48] was developed to secure V2X. The methodology of the NDN cloud is summarized in Table 1.

## 3. Mobility Methodology

KITE, the PMSS, and hybrid NeMo were selected due to the mobility support offered and the need for producer mobility technology. KITE uses the indirection method to transmit content using simple NDN communication; the PMSS improves producer operation by reducing handover latency; and hybrid NeMO provides a BIT to replace the FIB function by using the same cluster for indirection mobility. Thus, these three methods are comparable in measuring network performance for handover latency and signaling cost. On the basis of the NDN architecture, the ndnSIM simulator can be implemented on the network protocol stack framework. The simulator works with any link layer protocol, including point-to-point, CSMA, and wireless, as well as with the deployment of IPv4 and IPv6. The advantage of the ndnSIM simulator is that it can implement heterogeneous scenarios, such as NDN over IP. The ndnSIM simulator uses the C++ programming language to enable several modules in the NDN to operate, including PIT, FIB, content storage, network and application surface, and forwarding strategies. The use of this structured module enables any component to be modified.

NS3 has been patented, using the GNU GPLv2 license to reduce the writing of model procedures in simulations. The NS3 simulator uses the C++ library as its main program, as no graphics version is provided. NS3 can be used on Windows operating systems by adding MinGW modules. To capture the traffic log, the Wireshark app can be combined with the NS3 simulator. Several methods are running for the NDN testbed, especially on mobility perspectives, including KITE with C-IoT, PMSS with C-IoT, and hybrid NeMO with C-IoT, which is the latest technology. Figure 4 shows the setup for these three methods, which include a home router and a destination router; an AP that maintains the connection for mobile producers; and interaction between consumers with FIB to receive I_packets and D_packets.

Simulation is applied by researchers to implement new solution methods before testing them in real situations. Simulation methods can improve performance and meet performance expectations. Simulation combines elements of hardware and software. The combination of these elements is important, because running the simulation process can identify how some elements can communicate with each other and also identify the effect of the use of certain elements on the simulation process [59]. The network simulation process is unavoidable in the research article process, due to the use of open source model systems. The advantage of using open source systems is that researchers can conduct studies and find faults while the simulation process is in progress. With the use of network simulation using open source methods, various applications with different protocols can be run, such as FTP, TFTP, and Telnet. Network simulators can implement the methodological process for a study without implementing it in real situations. They can be implemented in different network topologies.

Various types of network simulators are available in today’s market. Different functions of each network simulator are used, depending on the accuracy of the parameters, i.e., the difficulty level, the number of nodes, the type of traffic on the nodes, the level of CPU usage, the memory usage, and the time rate calculation for the protocol used for a study. Several studies have been conducted on network simulators, such as NS3 [60], OMNET ++ [61], QualNet [62], and NetSIM [63]. Some articles on NDN simulators start with a basic setup to design a testbed for a preliminary test of NS3 [64].

### 3.1. KITE of NDN with C-IoT and AI

KITE technology with cloud operation uses a server called the rendezvous server (RS), which is located between the consumer and the mobility producer that uses IoT devices. The RS can always be traced by an NDN producer known as KITE. As shown in Figure 5, KITE operates between transmission I_packets from the consumer via accessing the producer by storing the location through the RS. A single route prefix is assumed for simplicity. KITE swaps the public keys to create confidence. It is signed by the mobile producer and has a special trace tag in the name, containing AI technology. The RS acknowledges the data interest for the location of the producer and returns signed tracing data to the mobile producer via the same path as the TI. Intermediary forwarders establish or update FIB entries for the mobile producer data prefix. When a mobile producer moves to other locations, it stores all of the prefixes via cloud storage for any interest packet update. Table 2 explains the operation of KITE technology in cloud storage.

### 3.2. The PMSS of NDN with C-IoT and AI

Several processes take place while using the PMSS with cloud to enhance producer mobility, as shown in Figure 6. The SR element is added to maintain data packet transmission with add-on cloud storage for IoT implementation and AI technology for processing data at mobile producers that use IoT devices as a transmission mechanism. Table 3 shows the details of the transmission of the PMSS.

### 3.3. Hybrid of NDN with C-IoT and AI

For hybrid NeMO with cloud operation, as shown in Figure 7, the mobile producer moves from the current location under agent router 1 (AR 1) to the newest location on AR 2. The processing of data in the mobile producer using this hybrid technology uses AI to speed up the transmission process. During this movement, all information of the data packet and the interest packet is currently updated through FIB, but in this technology, BIT reduces the functionality of FIB. AR 1 and AR 2 interact with each other on the NDN network cloud. After BIT has been updated at AR 2, it sends the latest information to MR 2 and, finally, to the mobile producer. After this operation, which is called interNeMO, the consumer obtains the data packet successfully. Table 4 explains the operation of hybrid NeMo in cloud storage.

## 4. Performance Comparison

As discussed in Section 3, each technology has its own advantages and a variety of operations. All this technology varies their levels of network performance, such as handover latency, signaling cost, and overhead cost.

This section will focus on handover latency and signaling cost on the perspective mobility of NDN. Each operation producer has three well-known methods for mobility: KITE, which is based on indirection technology; the PMSS, which is an evolution of the KITE operation; and hybrid NeMO, which is the newest method for NDN mobility. Handover delay, i.e., the time taken by a mobile producer that has received I_packets before disconnecting from their current connection, continues as new I_packets arrive at a new location. Handover latency is good when the value is smaller during the content transmission. Signaling cost, i.e., the measurement of the number of transmission messages from the node producer to the node consumer from NDN network communication, is good when the value is smaller while the content transmission is in progress.

To confirm this statement, a mathematical equation for KITE, the PMSS, and hybrid NeMO was designed to generate a formula for each network’s performance. For network analysis, ndnSIM was used to compare the performance evaluation for each method. For further setup in ndnSIM, several parameters were defined for each method to measure the performance and impact of mobility on each producer mobility technique. This setup measured the quality of network performance via each method by varying the number of parameters. Each topology used the same network grid of 400 m × 400 m; the range from the NDN router to AP was 100 m; the number of producers and mobility speeds varied for each method; the I_packets’ transmission range used 100 ms; and the same benchmark reference as that of the random waypoint mobility model was applied. Table 5 shows the details of how the parameters were set, including the size of the network, the speed of mobility, the mobility model used, and the simulation software on each testbed for different NDN mobility perspectives. Figure 8 shows the number of nodes for the consumer and the producer, which was used to measure the handover latency and signaling cost.

### 4.1. Signaling Cost

The signaling cost is the total traffic demand encountered while transmitting signaling messages during content transmission. According to [65], the handover rate is defined as the likelihood that the user crosses over to the next cell within one movement period. For this purpose, we focused on several messages that can be sent through the network during handover transmission. A low signaling cost corresponds to better performance of the proposed methodology.


**Signaling cost for KITE operation**


Equation (1) describes the signaling cost for the KITE operation, while Table 6 shows the parameter setup.
SC_Kite_ = C_prodAR_ + C_accessrouter_ + C_accessrouter_ + C_prodAR_ + C_consumerAR_ + C_accessrouter_ + C_accessrouter_+ C_prodAR_= S_Interest/data_ × (2a + 2c) + S_Interest_ × (2a + 2c)(1)


**Signaling cost for PMSS operation**


Equation (2) describes the signaling cost for the PMSS operation, while Table 6 shows the parameter setup.
SC_PMSS_ = C_prodAR_ + C_accessrouter_ + C_consumerAR_ + C_accessrouter_ + C_accessrouter_+ C_prodAR_= S_mobilityInterest_ × (a + 2c) + S_Interest_ × (a + c)(2)


**Signaling cost for hybrid NeMO operation**


Equation (3) describes the signaling cost for the hybrid NeMO operation, while Table 6 shows the parameter setup.
SC_hybrid_ = C_prodAR_ + C_accessrouter_ + C_consumerAR_ + C_prodAR_= S_mobilityInterest_ × (a + c) + S_Interest_ × (a + c)(3)


**Average signaling cost for each mobility operation**


Equation (4) describes the average signaling cost for each method.
SC_AVERAGE_ = (C_10_ + C_20_ + C_30_ + C_40_ +C_50_ + C_60_ +C_70_ + C_80_ + C_90_ + C_100_)/10(4)

### 4.2. Handover Latency

Handover, or handoff, is a method of mobile communication in which cellular transmission (voice or data) is transferred from one base station (cell site) to another without compromising the cellular transmission link. Handover is an important part of mobile transmission because it sets up data sessions or phone calls between always-mobile devices. Handover latency and failure are major problems in mobile data networks that slow down and stop service. We need lower latency to achieve better performance. Equation (5) refers to [66] to measure the speed of latency by calculating the average hop count of content transmission with a variety of mobile producer speeds.


**Handover latency equation for KITE operation**


Equation (5) describes the handover latency for the KITE operation, while Table 6 shows the parameter setup.
*HL_kite_* = ((*T_p_T_p_* + (*l*/*v*)) + (1 − (*T_p_T_p_* + (*l*/*v*)))) × (*l_pn_* + *a* × *Lwlup* + 2*d* × *LwtrcgInt* + *a* × *LwltrcdInt* + 2*d* × *LwtrcdIntA2*)(5)


**Handover latency equation for PMSS operation**


Equation (6) describes the handover latency for the PMSS operation, while Table 6 shows the parameter setup.
*HL_pmss_* = ((*T_p_T_p_* + (*l*/*v*)) + (1 − (*T_p_T_p_* + (*l*/*v*)))) × (*l_pn_* + *a* × *Lwl_mobInt_* + 2*c* × *Lwl_mobInt_* + *a* × *LWl_int_* + *c* × *LW_int_A*_2_)(6)


**Handover latency equation for hybrid NeMO operation**


Equation (7) describes the handover latency for hybrid NeMO operation, while Table 6 shows the parameter setup.
*HL*Hybrid = ((*T_p_T_p_* + (*l*/*v*)) + (1 − (*T_p_T_p_* + (*l*/*v*)))) × (*L_pn_* + *a* × *Lwl*_hybrid_ + 2*c* × *Lwl*_hybrid_ + *a* × *Lwl_int_* + *c* × *Lw_int_A*_2_)(7)


**Average handover latency equation for each mobility operation**


Equation (8) describes the average handover latency for each method.
HL_AVERAGE_ = (HL_10_ + HL_20_ + HL_30_ + HL_40_ + HL_50_ + HL_60_ + HL_70_ + HL_80_ + HL_90_ + HL_100_)/10(8)

## 5. Discussion and Analysis

This section discusses the performance evaluation for KITE, the PMSS, and hybrid NeMO, based on an IoT element embedded in the technology. Two types of performance evaluation were compared: signaling cost and handover latency.

### 5.1. Signaling Cost

Figure 9 shows how the signaling cost varied with mobile producer speed, starting from 0 ms to 100 ms after the simulation was run using the NS3 software based on the parameter sets.

The results were encouraging, despite the broadcasting nature of the three methods for the mobile producer and the domain restriction application. KITE and the PMSS had minimal handoff signaling costs. However, the handoff signaling cost of hybrid NeMO was nearly equal to or less than that of KITE and the PMSS, with about 4% of signaling cost, compared with those of KITE and the PMSS. The result of hybrid NeMO was good, even though no significant improvement against KITE and the PMSS was found at certain mobile producer speeds, such as 30 to 50 m/s, 50 to 60 m/s, and 70 to 100 m/s.

With regard to hop-count performance, Figure 10 shows that the KITE operation sent a greater number of messages than the PMSS or hybrid NeMO did. When the speed increased, the hybrid operation increased the number of messages to be sent. Therefore, mobility speed did not affect the performance of the transmission data packet and the interest packet to the consumer, because of the stability of BIT. Message drop was also encountered, but message delay generally happened during transmission.

Figure 10 shows that the hybrid method had the highest number of transmitted messages, compared with the PMSS and KITE. Even though a recent study stated that the PMSS and KITE have minimal handover signaling overhead during transmission, compared with anchor and anchorless producer mobility methods, the performance of the hybrid method in terms of signaling cost was more efficient than that of these other two methods.

### 5.2. Handover Latency

Figure 11 shows the handover latency versus variation in mobile producer speed from 0 ms to 100 ms after the simulation was run using the NS3 software based on the parameter set.

The handover latency of all schemes continued to increase, due to the increase in interest arrival rate. In addition, for the hybrid NeMO scheme, the handover latency was higher than that of KITE and the PMSS, because of the fast handover messages sent by the mobile producer as a pre-process of handoff. Moreover, the signaling cost of KITE was between that of hybrid NeMO and the PMSS, while hybrid NeMo had the lowest handover latency of 46% compared with KITE and 60% compared with the PMSS.

On the basis of a mathematical equations that were constructed for cost signaling and handover latency, a simulation was conducted by using the NS3 simulator to generate network performance with the use of a certain parameter. The techniques were compared to determine the hop count over speed. Figure 12 shows that while the mobile process increased the speed, the hop count to reach data for hybrid NeMo was less than that for the PMSS and KITE, because BIT decreased the usage of FIB, thereby reducing the hop count. This feature was an advantage of hybrid NeMO over the other two methods.

Handover latency can be measured by the time a mobile producer requests I_packets that were received from the last content router and the transmission of new I_packets after the handover operation. When more content needs to be traversed, the latency is increased because of the long distance from the consumer to the producer. The hop count between the consumer and the producer was examined. Figure 12 shows that when the speed of mobile producers increased, the hybrid method reduced the number of hops from the consumer to the producer. The average hop count for hybrid NeMo was, at most, 4 per hop, compared with the average hop count for the PMSS, which was 10 per hop; KITE had a maximum of 8 per hop while the mobile producer speed increased. Hybrid NeMO showed that having a lower average hop count could reduce handover latency, thus resolving the issues of path stretching for KITE.

## 6. Conclusions

This study adds comprehensively to current research on the merging of NDN, AI, and C-IoT technologies. This paper’s main contribution is the combination of NDN-AI-IoT. It focused on three selected methodological explanations for KITE, the PMSS, and hybrid NeMO. NDNs, rather than being simply an Internet technology, serve a variety of functions, with a focus on consumer-driven network architecture. KITE uses the indirection method to transmit content via simple NDN communication; the PMSS improves producer operations by lowering handover latency; and hybrid NeMO provides a binding information table to replace the base function of forwarding information. In addition, mathematical equations for signaling cost and handover latency were described in this study. The producer mobility operation was highlighted using the network simulator ndnSIM NS-3. Based on the mobility scenario, mathematical equations were developed for each methodology to measure handover latency and signaling cost. We concluded that the hybrid NeMO performs better than KITE and PMSS in terms of handover latency and signaling cost. In the future, this work will be expanded to include additional parameters with a larger network to support scalability and multiple environments.

## Figures and Tables

**Figure 1 sensors-22-06668-f001:**
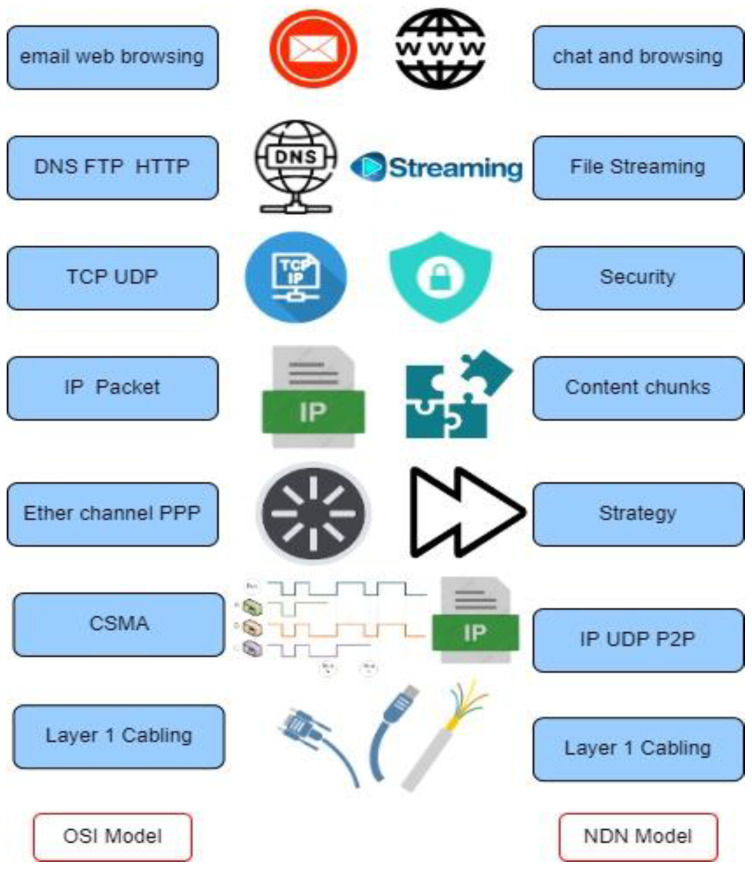
Comparison of the OSI model and the NDN model.

**Figure 2 sensors-22-06668-f002:**
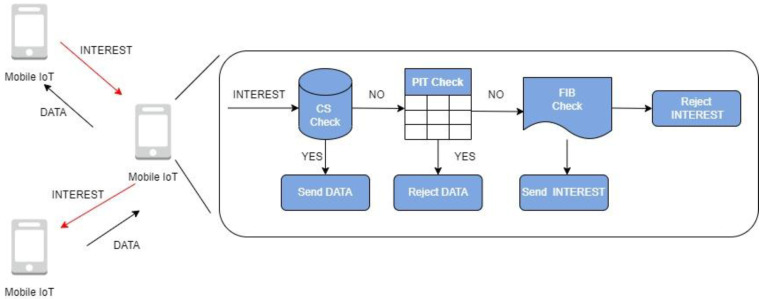
NDN forwarding strategies.

**Figure 3 sensors-22-06668-f003:**
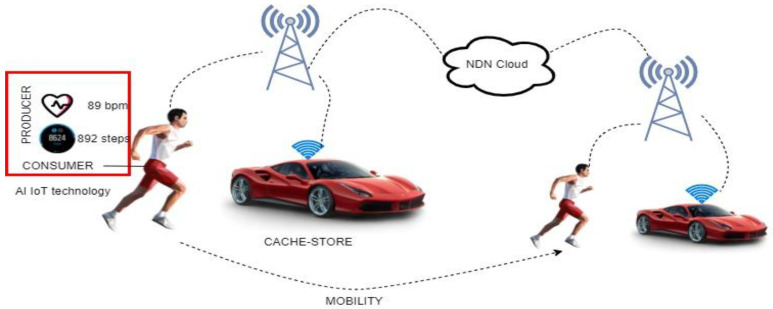
NDN AI mobility using smart technology.

**Figure 4 sensors-22-06668-f004:**
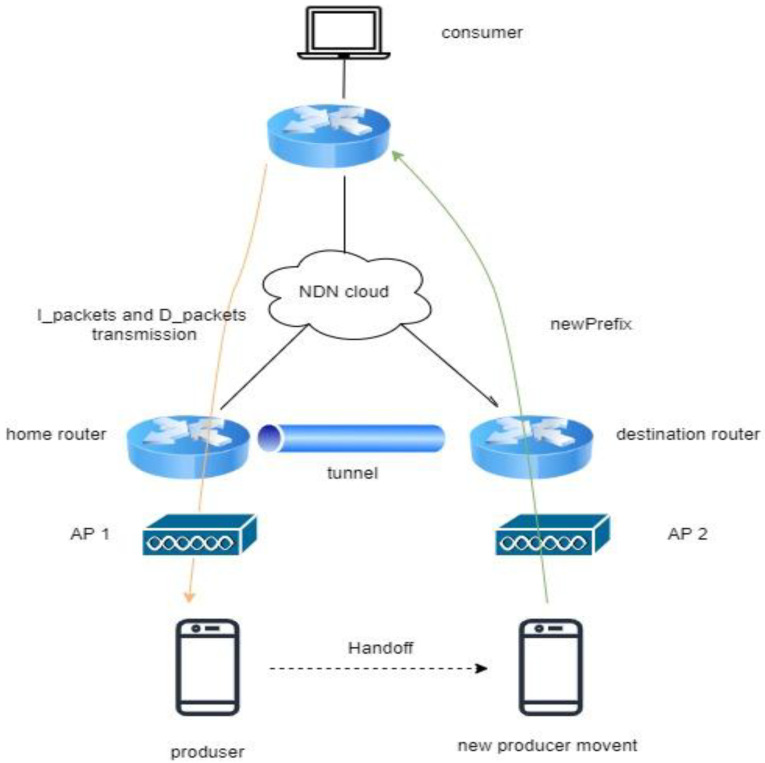
General NDN simulation scenario.

**Figure 5 sensors-22-06668-f005:**
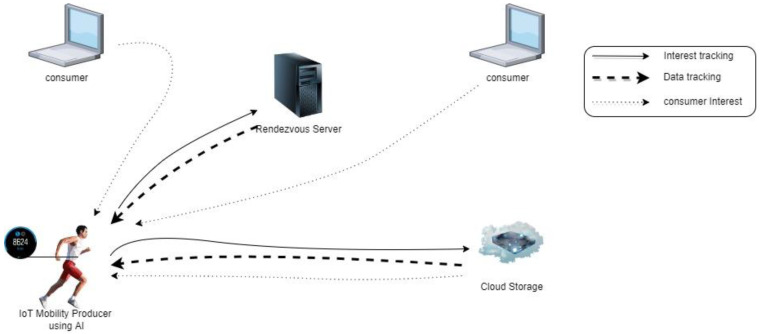
KITE operation with cloud storage.

**Figure 6 sensors-22-06668-f006:**
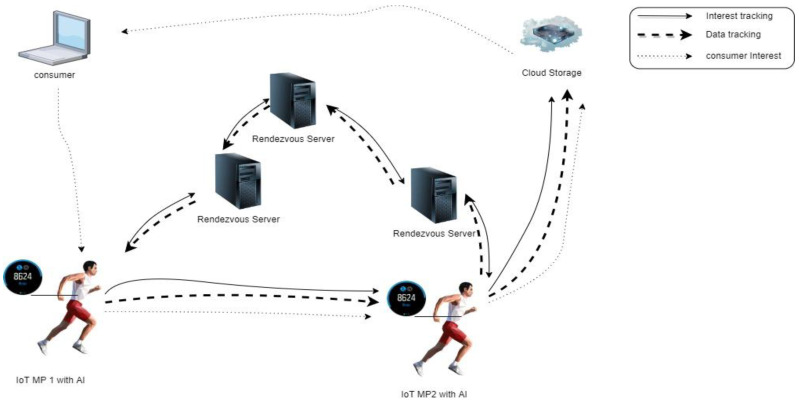
The PMSS with cloud operation.

**Figure 7 sensors-22-06668-f007:**
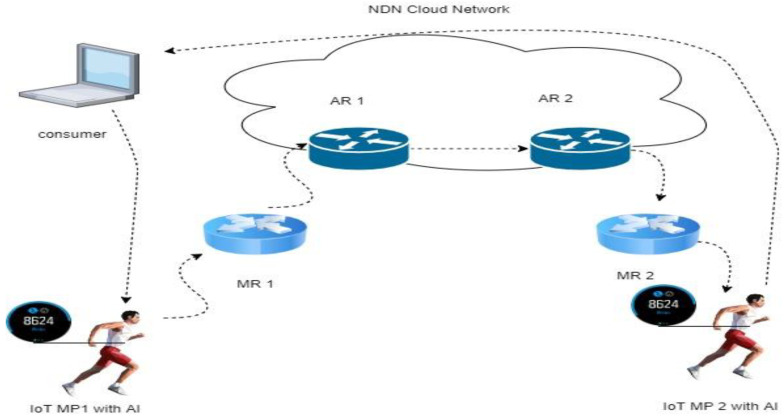
Hybrid NeMO with cloud operation.

**Figure 8 sensors-22-06668-f008:**
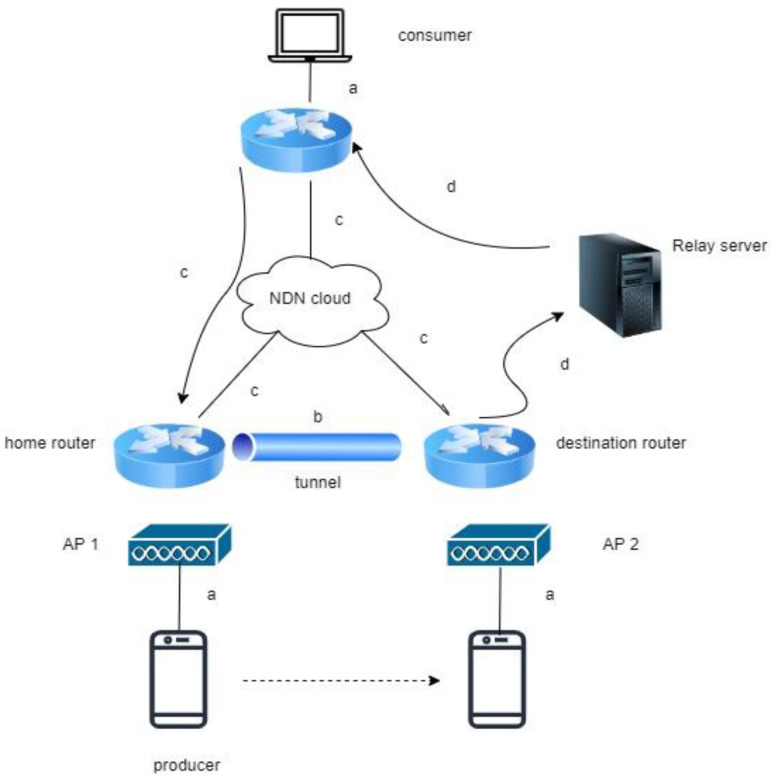
Network topology setup for KITE, the PMSS, and hybrid NeMO.

**Figure 9 sensors-22-06668-f009:**
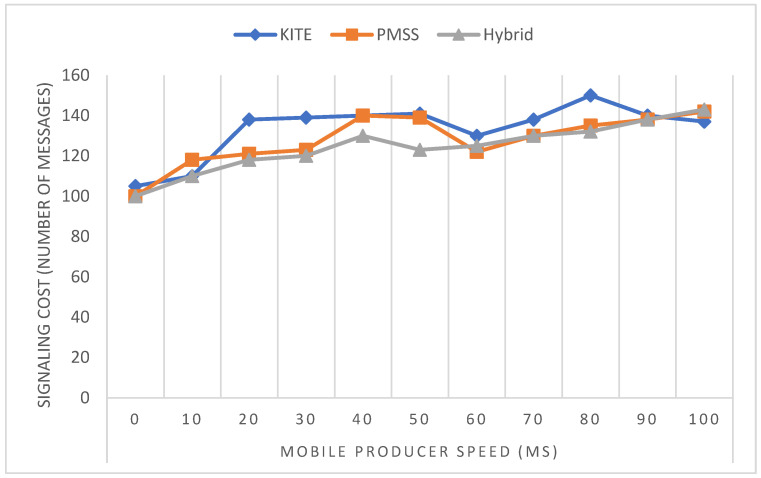
Signaling cost versus speed.

**Figure 10 sensors-22-06668-f010:**
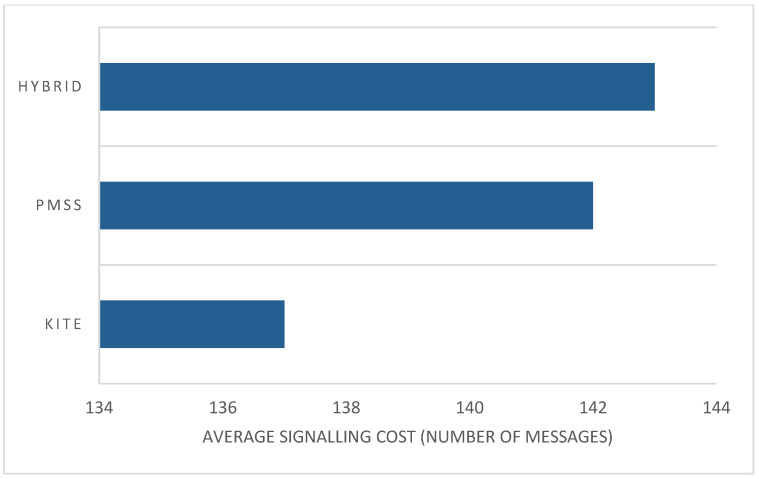
Average signaling by each method.

**Figure 11 sensors-22-06668-f011:**
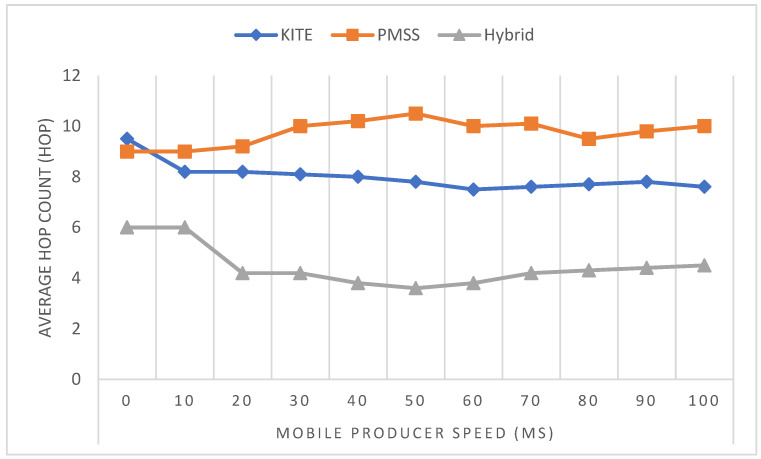
Handover latency (hop count) versus speed.

**Figure 12 sensors-22-06668-f012:**
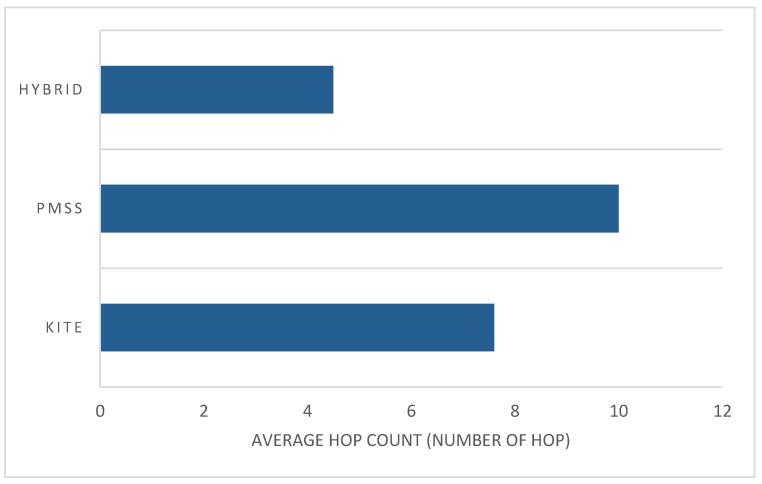
Average hop count by each method.

**Table 1 sensors-22-06668-t001:** NDN cloud application.

Reference	Method	Description
[49]	ERDOS	Integration of edge-native data flow and edge computing.
[50]	NDN-based aVC framework (NVCF)	Improve aVC data gathering success rates and lower the cost of aVC data retrieval.
[51]	NDN Genomics	With in-network caching of widely used datasets, NDN for genomic information operations improves data insights, accelerates extraction leveraging ubiquitous resources, and enables society to create architecture.
[52]	VC NDN	By using the NDN’s benefits, it provides cost-effective and value-based data retrieval.
[53]	NDN Routing overlay	In addition to reducing network and cache burden, NDN CSs that offer data aggregation and transformation also enforce privacy naturally.
[54]	VCCN	Vehicle networks have the potential to spread new data via multi-hop.
[55]	NDN IoT Edge	Cipher texts and signatures are used to ensure the security of medical data transport and the advantages of NDN.
[56]	VSN on NDN	Routing data between virtual sensors is a solution to the current paradigm.
[57]	NDN Mixed Reality Real-Time	Based on NDN, an AR/VR computational architecture that potentially addresses these issues is developed by utilizing a hybrid edge-cloud model.
[58]	NDN SGX-Based	Data access control keys are distributed and maintained efficiently and flexibly.

**Table 2 sensors-22-06668-t002:** KITE operation.

Flow of Transmission	Process Status	Description
Process 1	Before handoff	Typically, a consumer sends an I_packet over an NDN router to the network to request data. The NDN router then determines if the material is available; if not, then it sends an I_packet to the NDN network.
Process 2	Initiate connection with a content router	The I_packet’s prefix data name traverses NDN routers to reach the location of the producer. If the data are cached by any router in the network, then the router reacts promptly with the cached data. If not, then the routers, along with the path, store the interest information as entries in the PIT and FIB tables, then forward it until it reaches the producer. The producer then provides the data in breadcrumb format to the consumer.
Process 3	Handoff started	The producer abruptly decides to switch from the old PoA to the new PoA. After the connection, a new content name prefix is generated, and the producer prepares to notify the anchor router of the new name.
Process 4	Update RS	The producer overwhelms the network with trace I_packets destined for the immobile anchor router or the RS to notify it of the new name prefix.
Process 5	Establish data trace from producer to NDN router	Through intermediate routers, the anchor router or RS responds with trace data packets and establishes a trace between the mobile producer and the anchor router or RS.
Process 6	Store PIT in RS	The consumer saves PIT in the NDN router or RS.
Process 7	After the handoff operation	The NDN router forwards the consumer I_packet through data tracking at the mobile producer.
Process 8	Producer acknowledges	The mobile producer replies to the D_packet to the consumer through the NDN router.

**Table 3 sensors-22-06668-t003:** The PMSS’s operation.

Flow of Transmission	Process Status	Description
Process 1	Before transmission	Before handover transmission between the producer and the consumer, consumers transmit I_packets for requesting data to the mobile producer. The mobile producer checks the available content with SR1. If the content is not available, then it searches with other SRs in their neighborhood.
Process 2	Broadcast	Interest packet time is the time to retrieve the requested content and the time it takes to send it from the source to the destination.
Process 3	Processing data	The I_packet goes through the nearby RS until the mobile producer is reached. Processing data from the consumer to the mobile producer is similar to the processing of data in KITE operation behavior. While the handover process starts, the mobile producer cuts off the connection with RS1 and tries to find a greater signal with another RS from another zone. From the RS, a new naming prefix and a new mobile interest packet are created while reaching a new RS.
Process 4	FIB update	The next step is to broadcast the information to update the FIB that contains the routing information of the I_packet. Routing information is important to control the broadcast domain to make sure that no collision of I_packets occurs between the domain.
Process 5	Producer update	After this process is completed, the consumer resends a new interest packet to obtain new information from the producer. To maintain connectivity, NDN uses the best route strategy to reduce collision and forward the I_packet to the mobile producer’s new location.
Process 6	Cloud update	Cloud storage is used to update all transmission information and to store it in case the current connection is interrupted or fails. Thus, its advantage is that transmission is maintained and not disrupted.

**Table 4 sensors-22-06668-t004:** Hybrid Operation.

Flow of Transmission	Process Status	Description
Process 1	Exchanging information	Movement occurs from MR 1 to MR 2, and content is stored at MR2.
Process 2	Forwarding	MR 2 sends a signaling packet to AR 2 when AR 1 sends an alert on movement.
Process 3	Creating BIT	AR 2 and MR 2 create BIT for each entry of I_packets. BIT consists of information on consumer nodes, PoA, and face numbers.
Process 4	Matching BIT	BIT has a similar function to FIB, with I_packets searching for FIB to create a movement from the producer to the consumer.
Process 5	Forwarding I_packets	If the I_packet matches the information request by the consumer, then the content is directed without referring back to FIB.

**Table 5 sensors-22-06668-t005:** Parameter simulation setup.

Mobility Technique	Network Size (m^2^)	Distance Router NDN from AP	Mobile Producer Quantity	Mobility Speed (m/s)	Interest Range (ms)	Segment Size (bytes)	Mobility Model	Simulation Software	Benchmark Comparison
KITE	400 × 400	11 nodes/100 m	1	2	100	1024	Random waypoint mobility model	ndnSIM	KITE
PMSS	400 × 400	100 m	2	50, 200, 350 ms	100, 200, 300	1024	Random waypoint mobility model	ndnSIM	MBMA, CDBMA, CDBMA, IBMA
Hybrid NeMO	400 × 400	100 m	5	100 ms	100, 200	1024	Random waypoint mobility model	ndnSIM	KITE

**Table 6 sensors-22-06668-t006:** Network analysis parameter setup.

Parameter	Units	Description	Parameter/Value
S_data_	bytes	Size data packet	2000 bytes
S_Interest_	bytes	Size Interest packet	40 bytes
S_Interest_/S_mobilityInterest_	bytes	Size Interest packet	40 bytes
a	bytes	Packet transmission latency and cost between consumer and producer	1
c	bytes	Packet transmission latency and cost between the old NDN router and the new NDN router	5
*Tp*	ms	Paused time	0 ms, 100 ms
a	bytes	Packet transmission latency and cost between the consumer and the producer	1
*Lw*	ms	Wired link delay	2 ms
d	bytes	Packet transmission latency and cost between the consumer and the producer and the server	9

## Data Availability

Not applicable.

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
