# Peer review of "Comparison of Named Data Networking Mobility Methodology in a Merged Cloud Internet of Things and Artificial Intelligence Environment"

_sensors, 2022, doi:10.3390/s22176668_

Round 1
Reviewer 1 Report
This paper conducted a comparison study regarding NDN mobility methodology, which has strong practical background. The structure of the paper is well organdie. Before acceptance, some issues should be addressed.
1) Figure 1 in the introduction section is suggested to be moved in the background Section; Figure 2 in the introduction section is suggested to be deleted.
2) Regarding the recent advances in Internet of things, the work “Modeling and optimizing the cascading robustness of multisink wireless sensor networks”, “Modeling and analyzing cascading failures for Internet of Things.” can provide some help.
3) The performance comparison part is weak, I suggest the authors should enrich this part. Moreover, more details of the simulation setup should be presented.
4) For ease of understanding, Table 6, 7, 8 should be combined into one Table, similarly for Table 9, 10 and 11.
5) In the experiment part, the authors should present more discussions regarding the performance differences among three mobility model.
Reviewer 2 Report
The authors presented a Comparison of Named Data Networking Mobility Method ology in Cloud Internet of Things. The paper seams interesting and novel. However, I have following suggestions to improve its overall quality.
1. The article should be thoroughly revised for clarity and grammatical accuracy. In the present form, they are numerous grammatical errors which hamper its readability.
2. The abstract and conclusion should be improved to present the core contributions of the work and highlight the case study taken into account.
3. The Introduction section needs improvement. The authors should add the following subsections: Motivation to do the work, Contribution and Organization of the Paper..
4. Why these three methods are selected for comparison? The adaptation scenarios of each method are not fully described
5. In this paper, only the current three methods are compared, but there is no own innovation
7. Lack of optimization in routing.
8. in the Background section, Lack of relevant description under 5g/6G scenario, such as,
- Named Data Networking: A Survey. DOI: 10.1016/j.cosrev.2016.01.001
-Performance comparison between NEMO BSP and SINEMO. DOI: 10.1109/GLOCOM.2008.ECP.461
- A multiple-kernel clustering based intrusion detection scheme for 5G and IoT networks. Int. J. Mach. Learn. Cybern. 12(11): 3129-3144
- Building Agile and Resilient UAV Networks Based on SDN and Blockchain. IEEE Netw. 35(1): 57-63 (2021)
- Deep-Green: A Dispersed Energy-Efficiency Computing Paradigm for Green Industrial IoT. IEEE Trans. Green Commun. Netw. 5(2): 750-764 (2021)
-An Energy-Efficient In-Network Computing Paradigm for 6G. IEEE Trans. Green Commun. Netw. 5(4): 1722-1733 (2021)
Reviewer 3 Report
The paper presents NDN mobility challenges, focusing on Internet of Things (IoT) applications, cloud services, and artificial intelligence. The paper is organized well and gives a good introduction. But some changes are required to make the paper more readable. For instance, Fig. 1 is a comparison of OSI and NDN models. But it does not specify which side is OSI and which side is NDN in the figure. Quality of figures can also be improved to make them look sharp. Overall the paper is presented well.
Round 2
Reviewer 2 Report
The authors provides methodological and parameter setup explanations for KITE, producer mobility support scheme (PMSS) and hybrid network mobility (hybrid NEMO); constructs a mathematical equation for signalling cost and handover latency; and analyses network performance using NS3 simulations. The idea seams interesting and novel, however, I have the following suggestions to improve its overall quality.
1. The article should be thoroughly revised for clarity and grammatical accuracy. In the present form, they are numerous grammatical errors which hamper its readability.
2. The abstract and conclusion should be improved to present the core contributions of the work and highlight the case study of “Cloud IoT and AI” taken into account.
3. The threats of data security in named data network (NDN) should be discussed.
4. In terms of content storage, the setting of content cache needs further discussion.
5. The paper is full on the whole, but it is suggested to increase the discussion on the vulnerability of the proposed methods, such as the vulnerability in architecture and content storage.
6. The authors mention that they have highlighted the existing approaches similar to their work. However, some of the latest literature is not included, please add, such as,
- Building Agile and Resilient UAV Networks Based on SDN and Blockchain. IEEE Netw. 35(1): 57-63 (2021)
-A multiple-kernel clustering based intrusion detection scheme for 5G and IoT networks. Int. J. Mach. Learn. Cybern. 12(11): 3129-3144 (2021)
-An Energy-Efficient In-Network Computing Paradigm for 6G. IEEE Trans. Green Commun. Netw. 5(4): 1722-1733 (2021)
